# MALDI-TOF Mass Spectrometry for the Diagnosis of Citrus Canker Caused by *Xanthomonas citri* subsp. *citri*

**DOI:** 10.3390/molecules27248947

**Published:** 2022-12-15

**Authors:** Edenilson dos Santos Niculau, Douglas Ferreira, Edson Rodrigues-Filho, Franklin Behlau, Rodrigo Facchini Magnani, Leonardo Toffano, Evandro Luis Prieto, João Batista Fernandes, Maria Fátima das Graças Fernandes da Silva

**Affiliations:** 1Departamento de Química, Centro de Ciências Exatas e Tecnologia, Universidade Federal de São Carlos, Rod. Washington Luís, Km 235, São Carlos 13565-905, SP, Brazil; 2Centro de Ciências Integradas, Universidade Federal do Norte do Tocantins, Av. Paraguai, s/n—Esquina com Rua Uxiramas, Araguaína 77824-838, TO, Brazil; 3Fundo de Defesa da Citricultura—Fundecitrus, Av. Adhemar Pereira de Barros, 201, Araraquara 14807-040, SP, Brazil

**Keywords:** *Citrus sinensis*, plant, macromolecules, proteomic

## Abstract

Citrus canker, caused by the bacterium *Xanthomonas citri* subsp. *citri* (*Xcc*), is a disease that causes serious problems to the global citrus industry. Matrix-Assisted Laser Desorption/Ionization Time-of-Flight (MALDI-TOF) Mass Spectrometry (MS) has been used in human medicine to diagnose various diseases caused by both fungi and bacteria. In agriculture, this technique has potential for the diagnosis of diseases due to the low cost of large-scale analysis and quickness. This study showed that MALDI-TOF MS combined with chemometric analysis was effective for differentiating the macromolecule profile of orange leaves with canker lesions, healthy leaves, and leaves with phytotoxicity symptoms, proving that this technique may be used for the rapid diagnosis of citrus canker.

## 1. Introduction

Citrus is one of the fruit crops grown and produced in the world. China, Brazil, India, Mexico, and the United States are the biggest producers in the world. However, this crop is affected by a disease, citrus canker, that can harm the production, marketing, and consumption of this fruit [1]. Citrus canker (CC), caused by the bacterium *Xanthomonas citri* subsp. *citri* (*Xcc*) is an important disease for the citrus industry in several countries, such as Brazil and the United States, the two largest orange producers in the world. The bacterium affects all commercially important citrus varieties and species. The impacts of this disease are related to the defoliation of plants, the depreciation of the quality and reduction of production due to the presence of lesions in the fruit, premature drop of fruit, and the restriction for the commercialization to canker frees areas [1]. Symptoms have similar characteristics in leaves, fruit, and twigs. Lesions are necrotic, protruding, and brown. Often, the lesions are surrounded by yellowish chlorotic halo. The spread of the bacteria occurs mainly through rain splashes carried by wind, but also through people, seedlings, tools, and contaminated machines. When it reaches the plant surface, the pathogen can penetrate the host through stomata or wounds. The infection of citrus plants by *Xcc* occurs in young plant tissues such as shoots up to 4 to 6 weeks old and fruit up to 3 to 4 months old or 5 cm diameter [2]. Free water and temperature are important environmental factors for the occurrence of citrus canker [3]. Whereas water is important for the spread and penetration of the bacteria, temperature influences the multiplication of the bacteria inside the host and the development of lesions. Greater favorability to the pathogen occurs when the temperature is between 25 to 35 °C and the leaf wetness period is equal or greater than four consecutive hours. These are the optimal conditions for the bacteria to grow and spread in the plant. In vitro growth of the microorganism changes according to the composition of the culture medium, temperature, and humidity [3,4].

The disease occurs endemically in several citrus-producing regions such as Florida (USA), Argentina, Uruguay, and southern Brazilian states [1]. However, in several other important citrus-producing areas, the disease is not present, such as in Europe, South Africa, Mexico, California (USA), and in the states of Bahia and Sergipe, which make up the second largest Brazilian citrus production belt. In these areas, the early diagnosis of the disease for an immediate adoption of containment measures is critical for the suppression or eradication of the pathogen [1,5].

The main techniques used for the diagnosis of citrus canker are molecular PCR-based methods [6,7], serology [8], injection–infiltration bioassay or needle-prick pathogenicity tests [9,10], and culture-based methods [9,11,12,13]. PCR detects the bacterium with high precision and sensitivity by using its genetic code [6,7]. This method, however, is expensive and laborious and, hence, it is ill-suited for large-scale use. Furthermore, plants may contain some natural compounds that inhibit the amplification of nucleic acids in PCR [14,15], which may generate false-negative results in the diagnosis of the disease. Serological detection is based on the ability of an antibody to recognize and bind to a specific antigen, a substance associated with the plant pathogen. This method is known as enzyme-linked immunosorbent assay (ELISA). Regarding molecular diagnosis, the serological method is faster and more practical, as it can be performed in the field, but it requires a greater titer of the pathogen in the tissue sampled [16]. In turn, injection–infiltration and culture-based methods are less accurate and more time-consuming than other methods, but they do not require sophisticated equipment, material, or training for their application [13].

Spectroscopic and spectrometric methods appear to be promising alternatives for the diagnosis of many plant diseases, including citrus canker [17,18,19]. These techniques can rapidly measure the chemical and physical properties of the samples that are related to their chemical composition. One of these approaches is related to compounds produced by the plant defense mechanism (phytoanticepins) or to new compounds (phytoalexins), which can be secondary metabolites or specific macromolecules biosynthesized by interaction between the phytopathogen and host [20,21,22].

One way to detect the presence of macromolecules produced by the plant defense mechanism or by a plant–host interaction can be performed by Matrix-Assisted Laser Desorption/Ionization Time-of-Flight (MALDI-TOF) Mass Spectrometry. This technique has been demonstrated to be a useful tool in several applications, such as in clinical microbiology to identify pathogenic microorganisms, including fungi, yeast, and bacteria [23,24,25]. In agriculture, MALDI-TOF can also be an excellent means for diagnosing phytopathogens [26] due to the low cost of large-scale analyses, low sample consumption, high reproducibility, fast sample preparation, and rapid data acquisition. Despite the great potential, MALDI-MS has been restricted for investigation of the spatial distributions of metabolites in plant organs [27], as well as for taxonomic classification [28].

The aim of this study was to employ the MALDI-TOF MS technique associated with multivariate analysis as a potential tool for citrus canker diagnosis in infected *C. sinensis* leaves from the field.

## 2. Results and Discussion

The amplification of the 747 bp fragment, a specific fragment of the *Xcc* DNA region, referring to the atpD gene (F1-F0 subunit of ATP synthesis), confirmed the presence of the bacterium in the symptomatic leaves (Figure 1). Despite the accuracy and specificity, PCR and DNA sequencing analysis remains costly and time-consuming tools for plant pathogen diagnostics, reinforcing the need for the development of alternative or complementary techniques [29,30].

The analysis by MADI-TOF MS of *Xcc* (strain 306) allowed the identification of the bacteria at the genus level with high accuracy and at the species level with good probability in colonies from 48 h of cultivation (spotting and identifications in green) (Figure 2b and Figure 3b). In some repetitions (spots), the identification occurred only at the genus level with good probability (spots and identifications in yellow) (Figure 3b). When the bacterium was cultivated for 24 h, in only one repetition (spotting), it was possible to carry out the identification at the genus level with good probability (spotting in yellow) (Figure 3a), but in the other spots, it was not possible to accurately identify the bacterium at the genus and species level (spots and identifications in red) (Figure 2a and Figure 3a). There was a greater variation in ions detected between 24 and 96-h (Figure 2a and 2c, respectively). The ions detected at 96-h were the most discrepant compared to 24 and 48-h. Thus, the bacterium was not diagnosed with good probability in any of the 96-h repetition (spotting) (Figure 3c). The detection of ions with *m*/*z* 4664; 5859; 7418; and 9330 were important for the diagnosis in the battery in 48 h. These ions were also detected in the 24-h analysis; however, with small variations in the values. Some ions detected between *m*/*z* 4000 to 5000; 7000 to 7300; and 8700 to 10,000 were also relevant for the best classification of bacteria at the genus and species level in 48 h. They were not detected in 24 h and 96 h. This is justified probably by the fact that *Xcc* 306 reached the stationary phase of growth on NBY (Nutrient Broth Yeast) medium within 48 h of culture, so it led to a spectrum with many ions (Figure 2b); however, the optimal conditions for the in vitro growth of bacteria change according to the composition of the culture medium, temperature, and humidity. This hypothesis is based on the profile of macromolecules obtained by MALDI-TOF MS. Mass spectra are similar to those of bacteria in the software database (Figure 3b). The most characteristic ions of the bacterial macromolecules profile were detected in the range of *m*/*z* 5000 and *m*/*z* 7000. Sindt et al., 2017 also detected ions in this range in the analysis of some strains of this bacterium by MALDI-TOF MS [31].

The results of the chemometric analysis of samples healthy leaves tissue (HT), herbicide phytotoxicity leaves tissue (HP), infected leaves tissue (IT), and pure *Xcc* culture (PC) are represented by an MSP dendrogram (Figure 4). Three groups were formed at the 650-distance level: a group formed by samples containing IT samples, a mixed group formed by samples HT and HP, and another group formed only by PC.

Orange leaves can have necrotic symptoms caused by herbicide phytotoxicity that are similar to citrus canker, being very difficult to distinguish visually. Thus, the profile of macromolecules in leaves in the presence of herbicides was also extracted and evaluated. According to the dendrogram, there was no statistical difference between HT and HP, but all HP samples were separated from IT. This is likely because the profile of macromolecules extracted and detected in HP and IT differ from each other. The presence of the bacteria in IT causes a response in the plant; for example, a defense mechanism, producing phytoalexins or phytoanticipins [32], which, in the case of HP, may not be present, as they are not the same or change sharply in concentration. All PC samples differed from plant tissues (IT, HT, and HP), resulting in a separated group. This is likely due to the high similarity in the macromolecules profile considering only pure cultures and the high differentiation compared to plant samples.

Three ions detected in PC (*m*/*z* 3186.878; 4418.330; and 7327.092) (Figure 5a) and IT (*m*/*z* 3191.843; 4405.716; and 7315.839) (Figure 5b) are important to detect the presence of bacteria in the plant without using multivariate analysis; however, there are variations in the *m*/*z* values due to the low intensity of these ions in IT. Thus, it is recommended to perform a multivariate analysis similar to this work for the diagnoses of the microorganism and to separate the groups.

These results show that the multivariate analysis using the equipment software clearly differentiates the leaves containing the symptoms of citrus canker (IT) from the healthy leaves (HT) in 100% of the treatments. In other words, the technique is capable of detecting the *Xcc* + plant interaction, separating it from isolated samples of *Xcc* and healthy leaf tissues or with herbicide phytotoxicity lesions.

PCA clustering analysis (Figure 6) applied to our data shows that infected tissue samples (positive quadrant of PC1) were separated from the health tissue samples (negative quadrant of PC1), corroborating the results found in the dendrograms. Figure 7 shows the accumulated variance in each principal component, but the first two components represent most of the variance in the data.

The MALDI Biotyper PCA allows reducing the dimensionality of a data set in which there are a large number of interrelated variables, while retaining the variation. The data set consists of mass spectra obtained from different species. In mass spectra, the intensities of individual *m*/*z* ratios represent the variables. PCA calculates a new coordinate system that can be expressed as the linear combination of the original variables (mass-to-charge ratios *m*/*z*). This enables major trends in the data to be identified. PCA is based on eigenvalue/eigenvector decomposition of the covariance or the correlation matrix of the original variables. Data can usually be adequately described using far fewer coordinates (principal components) than original variables.

The loading plot for principal component 1 and principal component 2 showed that ions with *m*/*z* above 7000, framed in the positive quadrant of load1, have the highest weights for separating infected samples from healthy samples (Figure 8). Therefore, ions above this range can be used as disease markers. In fact, this range of ions was also detected in the analysis of bacterial cultures (Figure 2b).

As shown in Figure 9, a consistent reproducibility of mass spectra was obtained for the different treatments. This shows that the technique, together with the extraction procedures, is statistically favorable for use in many samples. There was also differentiation in the ions detected when they were compared between treatments (IT, HT, HP, and PC), with greater similarity between HP and HT and greater dissimilarity between IT and PC, corroborating the MSP dendrogram shown in Figure 4.

MALDI-TOF has already been used for diagnosis of diseases caused by microorganisms [24,25,26], including two pathovars of *Xanthomonas oryzae* [33], but in agriculture, it is still little employed [34,35]. The cost of the analyses described in this work is very low for a laboratory that has the MALDI-TOF MS equipment. The routine reagents used are distilled water, ultrapure water, and matrix solution containing 20 mg/mL of α-cyano-4-hydroxycinnamic acid (CHCA) in acetonitrile 0.2% aqueous trifluoroacetic acid (1:1). For example, for each leaf sample in triplicate, only 1.5 mL of distilled water, 25 µL of ultrapure water, and 4.5 µL of matrix solution are used. Thus, the estimated cost per sample in triplicate is only US$ 0.12. Furthermore, the final diagnosis takes only 3 h for a plant sample in triplicate. The relative total time can be reduced if the experiment is done in large-scale. This is the first time that this technique has been applied for the diagnosis of citrus canker. These results suggest that the MALDI-TOF MS technique has potential for the rapid diagnosis of not only citrus canker, but also other bacterium-causing disease that affect crops.

## 3. Materials and Methods

### 3.1. Sampling Groups

The extraction of the macromolecules was performed in four groups of samples, being (i) leaves with citrus canker lesion (IT), (ii) necrotic lesions resulting from phytotoxicity due to the herbicide Paraquat (1,1’-dimethyl- 4,4’-bipyridinium, C_12_H_14_N_2_) on leaves (HP), (iii) healthy leaves (HT), and (iv) pure cultures of *Xcc* (PC) (Figure 10). Leaves of Valencia sweet orange (*Citrus sinensis*) trees with no symptoms, with canker lesions, or with phytotoxicity injuries were collected in a 6-year-old orchard commercial located at the municipality of Guairaçá, Paraná state, Brazil. Necrotic lesions caused by herbicide phytotoxicity were included in the study because Paraquat is often used in citrus orchards for the control of weeds, and the accidental lesions on citrus leaves are often confused with citrus canker lesions in visuals diagnostics.

For each group of plant samples, forty-four leaves were collected, submitted to the macromolecule extraction process, and analyzed by MALDI-TOF MS. All plant samples were submitted to the same procedure for macromolecule extraction, according to Section 3.2.1. Fifteen isolates of pure cultures of *Xcc* from the bacterial collection of Fundecitrus (Araraquara, São Paulo, Brazil) were also selected (Table 1). Prior to the extraction of macromolecules, isolates were grown on the NBY (Nutrient Broth Yeast) agar medium (containing per liter of water: 5.0 g phytone peptone, 3.0 g beef extract, 2.0 g yeast extract, 2.0 g K_2_HPO_4_, 0.5 g KH_2_PO_4_, and 16 g bacteriological agar) at 28 °C and a 12-h photoperiod. After 24 h, the macromolecules profile of each isolate was determined by MALDI-TOF MS. The macromolecules profile from the wild strain 306 (PC 306) was determined in 24, 48, and 96 h.

### 3.2. MALDI-TOF Mass Spectrometry Analysis

#### 3.2.1. Extraction, Sample Preparation of Plant, and Analysis by MALDI-TOF MS

Citrus leaves with healthy tissue (HT), herbicide phytotoxicity (HP), and *Xcc*-infected tissue (IT) with canker infection lesions were collected and kept at −80 °C until the MALDI-TOF MS analysis. Pure culture (PC) of *Xcc* bacterium from different orchards were grown according to Section 3.1 before analysis by MALDI-TOF MS. PC samples were analyzed in the equipment using a direct spotting (smear) of bacterial colonies after each cultivation time. For the extraction of macromolecules from the leaves’ plant, lesions were excised (≈0.8 cm × 0.4 cm) and transferred to a vial tube, where 1.5 mL distilled water was added. Tubes were stirred for 2 h at 900 rpm. The supernatant was removed after centrifugation at 12,000 rpm, 4 °C for 4 min. The precipitated pellet was resuspended into 25 microliters of ultrapure water. Two microliters of solutions were sampled onto each twenty-four target spot of a MTP 384 ground steel TF target plate (Bruker Daltonics, Bremen, Germany) and air-dried at room temperature. Colonies from each bacterium grown were smeared using a sterilized toothpick onto each twenty-four target spots. Spotted samples were overlaid by 1.5 µL of matrix solution and air dried at room temperature before analysis. The matrix solution contained 20 mg/mL of α-cyano-4-hydroxycinnamic acid (CHCA) in acetonitrile 0.2% aqueous trifluoroacetic acid (1:1), and was sonicated before use to ensure the solubilization. The Protein Calibration Standard I (~5000–17,500 Da, Bruker Daltonics, Billerica, MA, USA) was used for the internal calibration of the mass spectrometer.

MALDI-TOF analyses were performed in the linear mode acquiring positive-charged ions produced by using a Smartbeam-II laser technology with an adjustable repetition rate of up to 1 KHz (355 nm) mounted in a Autoflex Speed spectrometer (Bruker Daltonics, Bremen, Germany). Acquisition was controlled by the FlexControl software package (version 3.3, Bruker Daltonics, Billerica, MA, USA) with automatic data acquisition, and mass spectra were acquired in the mass range *m*/*z* 2–20 KDa. Steps of 250 laser shots were applied to random walk automatic location methods inside of each target spot. The intensity and resolution of the extracted mass spectra were evaluated according to the software fuzzy logic control system. One spectrum from each target spot was generated by adding 1500 satisfactory laser single-shot with adequate intensity and resolution. A total of 24 spectra were acquired for each sample.

### 3.3. Construction of Main Spectra (MSPs)

The 24 spectra acquired to each sample were loaded into the MALDI Biotyper 3.1 software (Bruker Daltonics, Billerica, MA, USA) and preprocessed using standard algorithms for baseline subtraction, smoothing, and normalization described in the Biotyper Preprocessing Standard Method manual. Mass peaks in the range of *m*/*z* 3–15 KDa were used to compose a unified Main Spectrum (MSP) of each leaf sample.

The MSPs were created by using the Biotyper MSP Creation Standard Method of MALDI Biotyper 3.1 software. A peak list containing information on the average *m*/*z* and intensity of a maximum of 70 peaks present in at least 25% of the 24 spectra recorded for each sample was composed.

### 3.4. Statistical Analysis

The created MSPs were subjected to the Biotyper MSP Dendrogam Creation Standard Method for MSP Dendrogram construction.

The clustering PCA was created using the Standard Method of MALDI Biotyper 3.1 software with the number of principal components defined as automatic and following the mass parameters: lower bound (3000); upper bound (15,000); resolution (2).

### 3.5. Diagnosis of Citrus Canker by Polymerase Chain Reaction (PCR)

Definitive confirmation of the presence of the bacteria in canker-like lesions on citrus leaves was performed by the Polymerase Chain Reaction (PCR) technique. A specific region of the leaves from healthy and symptomatic leaves was cut for the analysis and extraction of total genomic DNA (plant + bacteria or plant only) with adaptations [36,37,38]. Using a porcelain pistil, the leaf sections were macerated in liquid nitrogen and aseptically transferred to 1.5 mL microtubes containing 600 μL of extraction buffer (NaCl 0.5 M; EDTA 0.05 M; Tris- 0.1 M HCl; pH 8.0; *β*-mercapoethanol 0.2%). After maceration, 50 μL of 20% SDS was added to each microtube, followed by shaking for one minute and incubation at 60 °C for 15 min. Subsequently, 300 μL of 5 M sodium acetate was added, stirred for one minute, and the obtained mixture was washed to centrifugation at 14000 rpm for 10 min. Then, 400 μL of the supernatant was transferred to new microtubes containing 400 μL of isopropanol. A brief agitation was performed in the microtubes and then they were taken to centrifugation at 14,000 rpm for 10 min. The supernatant was carefully removed from the microtube and 500 μL of 70% ethanol was introduced into each tube along with the precipitate. The microtubes were subjected to further centrifugation at 14000 rpm for 5 min. Finally, all the supernatant was discarded and the precipitate was dried at room temperature and resuspended into 50 μL of autoclaved Milli Q water to obtain the bacterial DNA.

PCR (Polymerase Chain Reaction) reactions were performed to amplify the atpD gene (F1-F0 subunit of ATP synthesis) “housekeeping genes”. Amplifications of the atpD gene were conducted with the primers (Primers), 5’-GGG CAA GAT CGT TCA GAT-3’ and 5’-GCT CTT GGT CGA GGT GAT-3’ [39]. Reactions were performed in 50 μL (final volume), with 1 U of high-fidelity DNA polymerase enzyme (Phusion, Finnzymes), 1X enzyme buffer, 0.2 mM dNTPs, 0.5 μM of each primer, and 50 ng of genomic DNA. The program used for the amplifications was carried out following the programming: 98 °C/30 s; 35 cycles at 98 °C/10 s, 68 °C/15 s, and 72 °C/15 s; and 72 °C/10 min. All amplifications were performed in an Eppendorf thermocycler model Mastercycler. The products generated were analyzed in 1% agarose gel, stained with ethidium bromide (0.1 pg/mL), visualized under ultraviolet light, and photo-documented.

## 4. Conclusions

MALDI-TOF MS proved to be a promising technique for the diagnosis of citrus canker through chemometric analysis. Furthermore, it is possible to differentiate plants with symptoms of herbicide application from canker disease. It is noteworthy that the methodology described in this work aids in visual diagnosis, as it is often difficult to diagnose the disease visually, given that the application of herbicides and other biotic and abiotic agents can also produce symptoms similar to those of citrus canker. Furthermore, due to the low cost and speed of large-scale analysis by MALDI-TOF MS, this technique has potential for application in the diagnosis of citrus canker. The methodology of this work is a breakthrough for the rapid diagnosis of citrus canker disease without isolation of bacterial DNA.

## Figures and Tables

**Figure 1 molecules-27-08947-f001:**
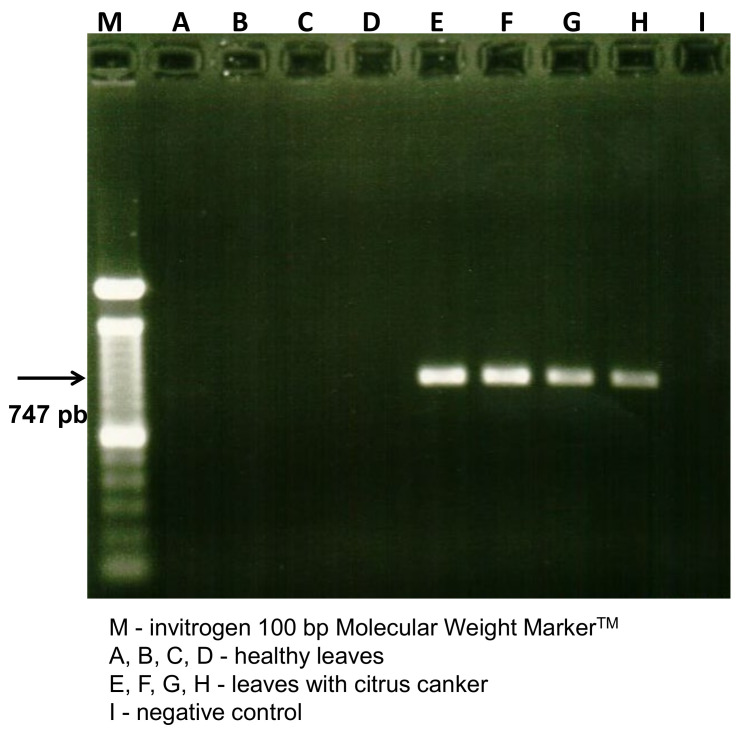
Diagnosis of citrus canker by PCR detection of *Xcc* in orange leaf samples characterized by the amplification of the 747 bp fragment. Four healthy leaves and four citrus canker leaves were used in the analysis.

**Figure 2 molecules-27-08947-f002:**
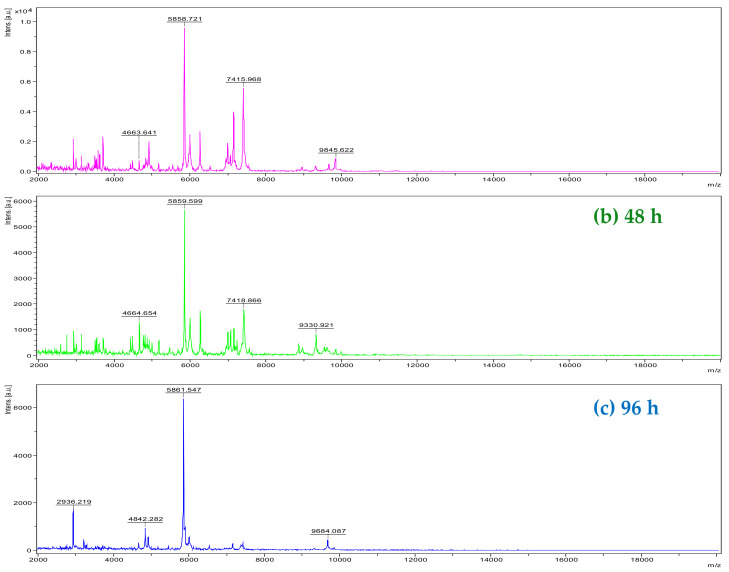
Profile of *Xcc* macromolecules by MALDI-TOF MS at 24 (**a**), 48 (**b**), and 96 h (**c**) after plating onto NBY agar medium. *y*-axis: ion intensity. *x*-axis: mass/charge (*m/z*).

**Figure 3 molecules-27-08947-f003:**
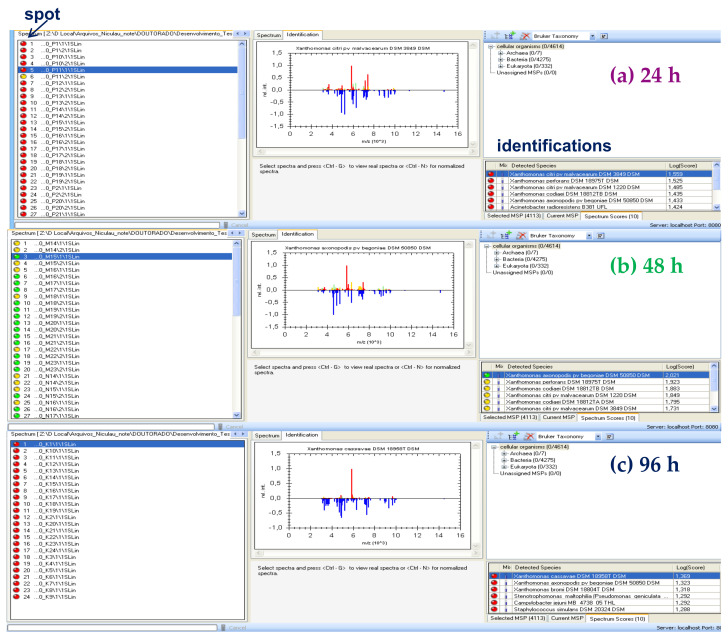
Diagnosis of *Xcc* by MALDI-TOF/MS using the software MALDI Biotyper after plating onto NBY agar medium in 24 (**a**), 48 (**b**), and 96 (**c**) h.

**Figure 4 molecules-27-08947-f004:**
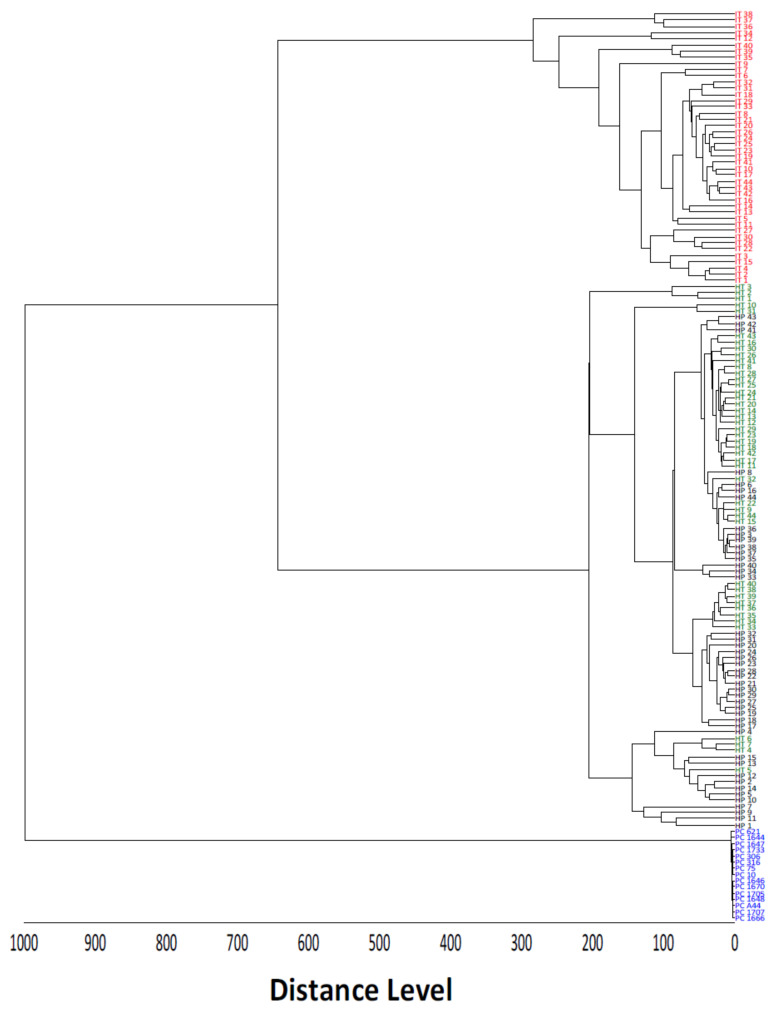
Maldi Biotyper MSP dendrogram of *C. sinensis* leaves samples from the field (IT, HT, and HP) and *Xcc* strains. IT: canker lesion, HT: healthy tissue, HP: herbicide phytotoxicity, PC: pure culture of *Xcc*.

**Figure 5 molecules-27-08947-f005:**
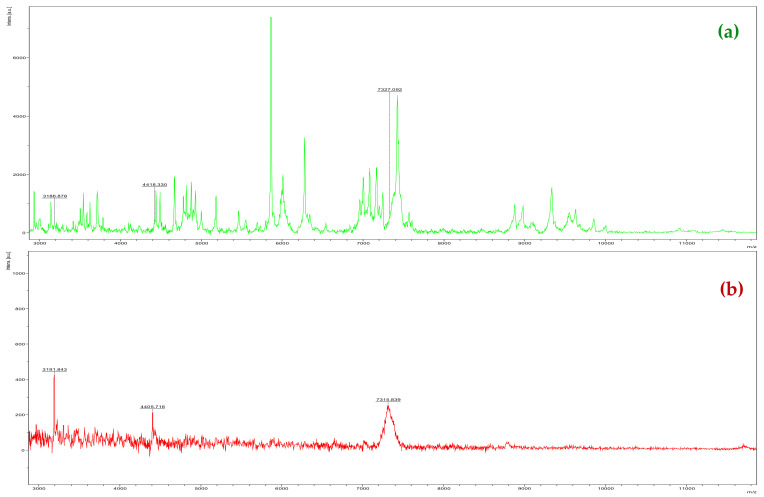
MALDI-TOF MS spectrum of pure culture (PC) of *Xcc* at 48 h after plating onto NBY agarmedium (**a**) and leaf sample with citrus canker (IT) (**b**). *y*-axis: ion intensity. *x*-axis: mass/charge (*m*/*z*).

**Figure 6 molecules-27-08947-f006:**
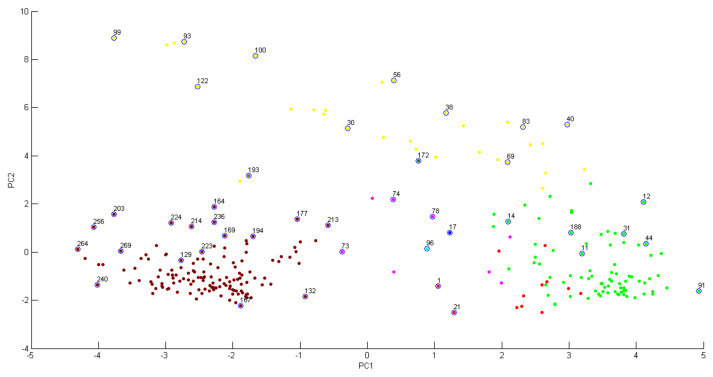
Maldi Biotyper PCA Clustering analysis of infected and healthy *C. sinensis* leaves samples. The numbers indicate the treatments of the plant tissues: 1 to 128 = IT; 129 a 270 = HT.

**Figure 7 molecules-27-08947-f007:**
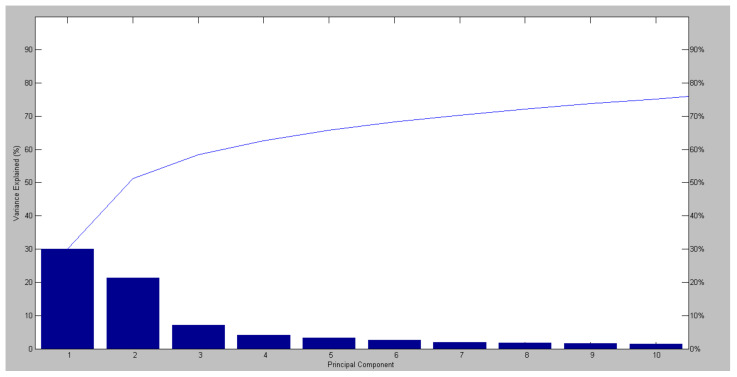
Variance explained from Maldi Biotyper PCA Clustering analysis of infected and healthy *C. sinensis* leaves samples. *y*-axis indicates the variance explained and *x*-axis indicates the principal component.

**Figure 8 molecules-27-08947-f008:**
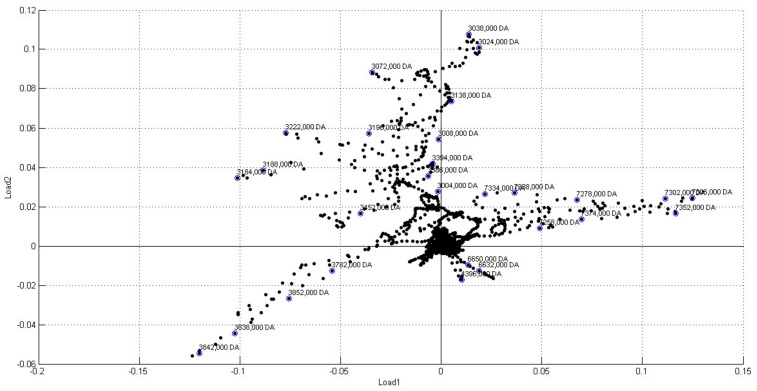
Load plot from Maldi Biotyper PCA Clustering analysis of infected and healthy *C. sinensis* leaves samples. The numbers indicate the ions detected.

**Figure 9 molecules-27-08947-f009:**
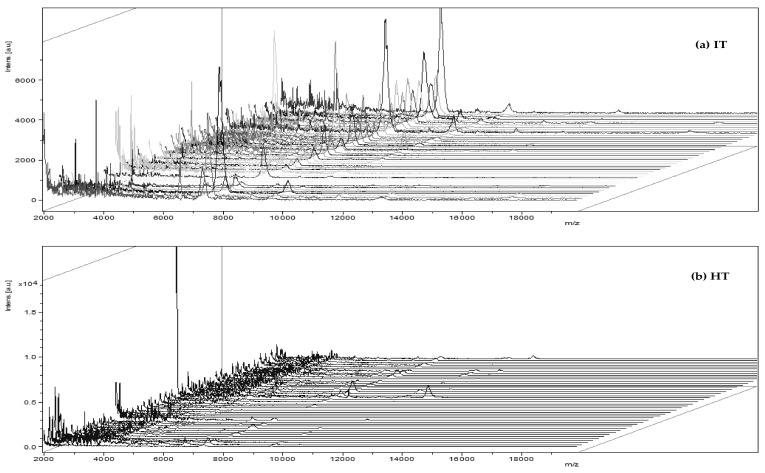
Mass spectra of *C. sinensis* leafsamples from the field, (**a**) IT, (**b**) HT, and (**c**) HP, and *Xcc* strains (**d**) PC. IT: canker lesion, HT: healthy tissue, HP: herbicide phytotoxicity, PC: pure culture of *Xcc*. *y*-axis: ion intensity. *x*-axis: mass/charge (*m/z*).

**Figure 10 molecules-27-08947-f010:**
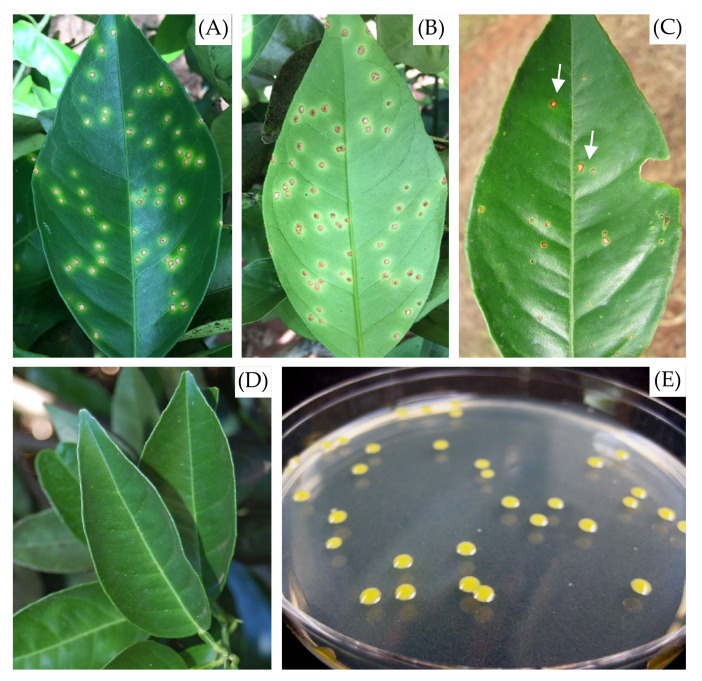
Treatments used to extract macromolecules for MALDI-TOF mass spectrophotometry analysis. Citrus canker lesions on the upper (**A**) and lower (**B**) surface of *Citrus sinensis* leaves: IT; necrotic lesions resulting from phytotoxicity due to the herbicide Paraquat (**C**): HP; healthy leaves (**D**): HT; and pure cultures of *Xcc* (**E**): PC. Arrows indicate necrotic lesions resulting from phytotoxicity due to the herbicide Paraquat.

**Table 1 molecules-27-08947-t001:** Code for mass spectrometry and strains of *Xcc*.

MS Code	Strain Identification	Origin	Year of Isolation
PC 1646	*Xcc*-03-1622	Bella Vista, Corrientes, Argentina	2003
PC 1644	1645	Tres Lagunas, Formosa, Argentina	2010
PC 1647	*Xcc*-07-3180	Monte Caseros, Corrientes, Argentina	2007
PC A44	A44	Bella Vista, Corrientes, Argentina	2003
PC 1648	FDC 1648	Santa Rosa, Corrientes, Argentina	2010
PC 1666	FDC 1666	Rondon, Paraná, Brazil	2011
PC 1670	FDC 1670	Alto Paraná, Paraná, Brazil	2011
PC 1705	FDC 1705	Paranavaí, Paraná, Brazil	2013
PC 1707	FDC 1707	Alto Paraná, Paraná, Brazil	2013
PC 1733	FDC 1733	Guairaçá, Paraná, Brazil	2014
PC 306	306	Paranavaí, Paraná, Brazil	1997
PC 75	FDC 75	Casa Branca, São Paulo, Brazil	1998
PC 316	FDC 316	Presidente Prudente, São Paulo, Brazil	2002
PC 621	FDC 621	Chapecó, Santa Catarina, Brazil	2001
PC 10	FDC 10	Guararapes, São Paulo, Brazil	1980

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
