# Peer review of "MALDI-TOF Mass Spectrometry for the Diagnosis of Citrus Canker Caused by Xanthomonas citri subsp. citri"

_molecules, 2022, doi:10.3390/molecules27248947_

Round 1

Reviewer 1 Report

The article entitled “MALDI-TOF mass spectrometry for the diagnosis of citrus canker caused by Xanthomonas citri subsp. citri” describe how MALDI-TOF MS is effective for differentiating macromolecule profile of orange leaves with canker lesions, healthy leaves and leaves with phytotoxicity symptoms. It was proved that this technique may be used for a rapid diagnosis of citrus canker. The article is well written, with some minor putative improvement, and quite interesting. The approach is pretty much like an advances in plant disease diagnosis and justifies the previous work reported in some recent studies but with some different interesting results. To improve the quality of the article I added below some suggestions that I hope might be useful.

Line 19: MALDI-TOF MS… write in full when use first time

Keywords: Don’t repeat the words used already in topic.

Line 42: citation in text format is not according to the journal’s instructions. Citation in numbering format should be followed.

Some references are too old. Similar information is available in recent studies. Please cite recent studies.

Introduction in general missing about the information and recent status of the citrus/host plant. Recent status of the citrus, its importance etc. should be introduced in introduction part

It is important to show the pictures of samples (leaves) collected for analysis

Line 202: “For each group of plant samples, forty-four leaves were collected, submitted to the macromolecule extraction process and analyzed by MALDI-TOF MS”….. weight of the sample is more good to describe instead of number of sample.

Line 206: NBY agar medium, write the full name of the medium

A separate paragraph should be used in material and method section for describing information of statistical analysis/test used.

What about the cost effectiveness of your proposed diagnostic tool? need to discuss in discussion part.

Although the results were discussed better. More improvement regarding relevant citation and depth of the study is needed.

It is important to add some key points of findings obtained in this study in conclusion section. 

Author Response

Response to Reviewer 1

The article entitled “MALDI-TOF mass spectrometry for the diagnosis of citrus canker caused by Xanthomonas citri subsp. citri” describe how MALDI-TOF MS is effective for differentiating macromolecule profile of orange leaves with canker lesions, healthy leaves and leaves with phytotoxicity symptoms. It was proved that this technique may be used for a rapid diagnosis of citrus canker. The article is well written, with some minor putative improvement, and quite interesting. The approach is pretty much like an advances in plant disease diagnosis and justifies the previous work reported in some recent studies but with some different interesting results. To improve the quality of the article I added below some suggestions that I hope might be useful.

Response: Thank you for reviewing and contributing to our work. We made corrections and improvements, according to the revised manuscript and descriptions below.

Line 19: MALDI-TOF MS… write in full when use first time

Response: This part was corrected (Line 18-19)

Keywords: Don’t repeat the words used already in topic.

Response: Some keywords have been replaced.

Line 42: citation in text format is not according to the journal’s instructions. Citation in numbering format should be followed.

Response: The reference has been corrected.

Some references are too old. Similar information is available in recent studies. Please cite recent studies.

Response:  Some reference was updated.

Introduction in general missing about the information and recent status of the citrus/host plant. Recent status of the citrus, its importance etc. should be introduced in introduction part

Response: The information was introduced.

It is important to show the pictures of samples (leaves) collected for analysis

Response: Figure 10 was inserted in the work.

Line 202: “For each group of plant samples, forty-four leaves were collected, submitted to the macromolecule extraction process and analyzed by MALDI-TOF MS”….. weight of the sample is more good to describe instead of number of sample.

Response: We did not weigh the leaves because we considered that the 0.8cm x 0.4 dimension is more relevant for the diagnosis, due to the symptom characteristics of the disease.

Line 206: NBY agar medium, write the full name of the medium

Response: full name of the medium was written (line 121)

A separate paragraph should be used in material and method section for describing information of statistical analysis/test used.

Response: This information was separated into a subsection.

What about the cost effectiveness of your proposed diagnostic tool? need to discuss in discussion part.

Response: A discussion was introduced (line 238 – 246)

Although the results were discussed better. More improvement regarding relevant citation and depth of the study is needed.

Response: The discussions were improved according to markings in the text.

It is important to add some key points of findings obtained in this study in conclusion section.

Response: More information was added in the conclusion

Reviewer 2 Report

Niculau et al., examined application of MALDI-TOF for identifying Xanthomonas citri. This method is widely used in medical microbiology and provided the resource availability, this could be a easy to use platform for bulk sample identification. Paper is well written, and my critics are listed below.

Line 44- Explain the optimum condition for the bacteria to grow.

Line 80- Remove virus as no clinical viral samples are ID using MALDI-TOF.

Fig.1 Suggest that 48 hrs has the best spectrum consistency. Expand the fig legends please. Mention how many samples were applied for representative figures and were there any discrepancies of the peaks on the same incubation periods? Explain in the results section.

Fig.4. IT and PC show distinct phylogeny. When you make fig 8, did you demarcate any common spectrum peaks between them? Those will be important for correct diagnosis. Explain in the results section.

Line 139-140- To prove your data, show spectrum similarities between IT and PC.

Add new fig with MALDI-TOF spectrum with saliant common peaks to ID citrus canker and pure cultures so that a technician without deep-core knowledge of proteomics and MALDI-TOF theory could be able to  ID samples and interpret. this would be the big picture and take-home message.

Line 212-225- When you use extraction protocol justification, subject pure culture with the same extraction protocol and show that there is absence or presence of spectral differences.

Author Response

Response to Reviewer 2

Niculau et al., examined application of MALDI-TOF for identifying Xanthomonas citri. This method is widely used in medical microbiology and provided the resource availability, this could be a easy to use platform for bulk sample identification. Paper is well written, and my critics are listed below.

Response: Thank you for reviewing and contributing to our work. We made corrections and improvements, according to the revised manuscript and descriptions below.

Line 44- Explain the optimum condition for the bacteria to grow.

Response: An information was introduced in the introduction (line 49-52)

Line 80- Remove virus as no clinical viral samples are ID using MALDI-TOF.

Response: The word “virus” was removed (line 86)

Fig.1 Suggest that 48 hrs has the best spectrum consistency. Expand the fig legends please. Mention how many samples were applied for representative figures and were there any discrepancies of the peaks on the same incubation periods? Explain in the results section.

Response: Text and legend have been updated.

Fig.4. IT and PC show distinct phylogeny. When you make fig 8, did you demarcate any common spectrum peaks between them? Those will be important for correct diagnosis. Explain in the results section.

Response:  Figure 5 was added and a discussion was introduced.

Line 139-140- To prove your data, show spectrum similarities between IT and PC.

Response: Figure 5 was added and a discussion was introduced.

Add new fig with MALDI-TOF spectrum with saliant common peaks to ID citrus canker and pure cultures so that a technician without deep-core knowledge of proteomics and MALDI-TOF theory could be able to  ID samples and interpret. this would be the big picture and take-home message.

Response:  Figure 5 and 10 were added and a discussion was introduced in the results and discussion.

Line 212-225- When you use extraction protocol justification, subject pure culture with the same extraction protocol and show that there is absence or presence of spectral differences.

Response: An information was introduced in section 3.1 (line 263-265)

Round 2

Reviewer 2 Report

Thanks for addressing all the concerns raised, except Fig.5 data. Please show matching spectrum peaks of PC and IT if you needed to correlate organism PC (fig.5A) and IT (Fig.5B)

present figure does not show common peaks.

Author Response

Response to Reviewer 2

Thanks for addressing all the concerns raised, except Fig.5 data. Please show matching spectrum peaks of PC and IT if you needed to correlate organism PC (fig.5A) and IT (Fig.5B)

Response: Thanks for reviewing and contributing to our work.  We made corrections showing the peaks of PC and IT (Line 159-160).